# The Many Faces of Optimal Weak-to-Strong Learning

**Mikael Møller Høgsgaard**
Department of Computer Science
Aarhus University
hogsgaards@cs.au.dk

**Kasper Green Larsen**
Department of Computer Science
Aarhus University
larsen@cs.au.dk

**Markus Engelund Mathiasen**
Department of Computer Science
Aarhus University
markusm@cs.au.dk

## Abstract

Boosting is an extremely successful idea, allowing one to combine multiple low accuracy classifiers into a much more accurate voting classifier. In this work, we present a new and surprisingly simple Boosting algorithm that obtains a provably optimal sample complexity. Sample optimal Boosting algorithms have only recently been developed, and our new algorithm has the fastest runtime among all such algorithms and is the simplest to describe: Partition your training data into 5 disjoint pieces of equal size, run AdaBoost on each, and combine the resulting classifiers via a majority vote. In addition to this theoretical contribution, we also perform the first empirical comparison of the proposed sample optimal Boosting algorithms. Our pilot empirical study suggests that our new algorithm might outperform previous algorithms on large data sets.

## 1 Introduction

Boosting is one of the most powerful machine learning ideas, allowing one to improve the accuracy of a simple base learning algorithm $\mathcal{A}$. The main idea in Boosting, is to iteratively invoke the base learning algorithm $\mathcal{A}$ on modified versions of a training data set. Each invocation of $\mathcal{A}$ returns a classifier, and these classifiers are finally combined via a majority vote or averaging. Variations of Boosting, including Gradient Boosting [10, 16, 7], are often among the best performing classifiers in practice, especially when data is tabular. Furthermore, when combined with decision trees or regressors as the base learning algorithm, these algorithms are independent of scaling of data features and provides impressive out-of-the-box performance. See the excellent survey [22] for further details.

The textbook Boosting algorithm for binary classification, AdaBoost [9], works by maintaining a weighing $D_t = (D_t(1), \ldots, D_t(m))$ of a training set $S = (x_1, y_1), \ldots, (x_m, y_m)$ with $(x_i, y_i) \in \mathcal{X} \times \{-1, 1\}$ for an input domain $\mathcal{X}$ and labels $\{-1, 1\}$. In each Boosting iteration $t$, a classifier $h_t : \mathcal{X} \to \{-1, 1\}$ is trained to minimize the 0/1-loss on $S$, but with samples weighed according to $D_t$. The weights are then updated such that samples $(x_i, y_i)$ misclassified by $h_t$ have a larger weight under $D_{t+1}$ and correctly classified samples have a smaller weight. Finally, after a sufficient number of iterations $T$, AdaBoost combines the classifiers $h_1, \ldots, h_T$ into a *voting classifier* $f(x) = \text{sign}(\sum_t \alpha_t h_t(x))$ taking a weighted majority vote among the predictions made by the $h_t$'s. Here the $\alpha_t$'s are real-valued weights depending on the accuracy of $h_t$ on $D_t$.

38th Conference on Neural Information Processing Systems (NeurIPS 2024).

**Weak-to-Strong Learning.** Boosting was originally introduced to address a theoretical question asked by Kearns and Valiant [17, 18] on *weak-to-strong* learning. A learning algorithm $\mathcal{A}$ is called a weak learner if, for *any* distribution $\mathcal{D}$ over $\mathcal{X} \times \{-1, 1\}$, when given some $m_0$ i.i.d. training samples $S$ from $\mathcal{D}$, it produces with probability at least $1 - \delta_0$, a classifier/hypothesis $h_S \in \mathcal{H}$ with $\mathrm{er}_{\mathcal{D}}(h_S) \leq 1/2 - \gamma$, where $\mathrm{er}_{\mathcal{D}}(h) = \mathrm{Pr}_{(x,y) \sim \mathcal{D}}[h(x) \neq y]$ and $\mathcal{H} \subseteq \mathcal{X} \to \{-1, 1\}$ is a predefined hypothesis set used by $\mathcal{A}$. A weak learner thus produces, for any distribution $\mathcal{D}$, a hypothesis that performs slightly better than random guessing when given enough samples from $\mathcal{D}$. The parameter $\gamma$ is called the *advantage* of the weak learner and we refer to the weak learner as a $\gamma$-weak learner. We think of $\delta_0$ and $m_0$ as constants that may depend on $\mathcal{H}$, but not $\mathcal{D}$. A strong learner in contrast, is an algorithm that for any distribution $\mathcal{D}$, and any parameters $(\varepsilon, \delta)$ with $0 < \varepsilon, \delta < 1$, when given $m(\varepsilon, \delta)$ i.i.d. samples $S$ from $\mathcal{D}$, produces with probability at least $1 - \delta$ a hypothesis $h_S : \mathcal{X} \to \{-1, 1\}$ with $\mathrm{er}_{\mathcal{D}}(h_S) \leq \varepsilon$. Here $m(\varepsilon, \delta)$ is the *sample complexity* of the strong learner. A strong learner thus obtains arbitrarily high accuracy when given enough training samples. With these definitions in place, Kearns and Valiant asked whether it is always possible to produce a strong learner from a weak learner. This was indeed shown to be the case [25], and AdaBoost is one among many examples of algorithms producing a strong learner from a weak learner.

**Sample Complexity.** Given that weak-to-strong learning is always possible, a natural question is "what is the best possible sample complexity $m(\varepsilon, \delta)$ of weak-to-strong learning?". This is known to depend on the VC-dimension $d$ of the hypothesis set $\mathcal{H}$ used by the weak learner, as well as the advantage $\gamma$ of the weak learner. In particular, the best known analysis [27] of AdaBoost shows that it achieves a sample complexity $m_{\mathrm{Ada}}(\varepsilon, \delta)$ of

$$m_{\mathrm{Ada}}(\varepsilon, \delta) = O\left(\frac{d \ln(1/(\varepsilon\gamma)) \ln(d/(\varepsilon\gamma))}{\gamma^2 \varepsilon} + \frac{\ln(1/\delta)}{\varepsilon}\right). \tag{1}$$

Larsen and Ritzert [20] were the first to give an algorithm improving over AdaBoost. Their algorithm has a sample complexity $m_{\mathrm{LR}}(\varepsilon, \delta)$ of

$$m_{\mathrm{LR}}(\varepsilon, \delta) = O\left(\frac{d}{\gamma^2 \varepsilon} + \frac{\ln(1/\delta)}{\varepsilon}\right).$$

They further complemented their algorithm with a lower bound proof showing that any weak-to-strong learning algorithm must have a sample complexity $m(\varepsilon, \delta)$ of

$$m(\varepsilon, \delta) = \Omega\left(\frac{d}{\gamma^2 \varepsilon} + \frac{\ln(1/\delta)}{\varepsilon}\right).$$

The optimal sample complexity for weak-to-strong learning is thus fully understood from a theoretical point of view.

**Other Performance Metrics.** Sample complexity is however not the only interesting performance metric of a weak-to-strong learner. Furthermore, $O(\cdot)$-notation may hide constants that are too large for practical purposes. It is thus worthwhile to develop alternative optimal weak-to-strong learners and compare their empirical performance.

The algorithm of Larsen and Ritzert for instance has a rather slow running time as it invokes the weak-learner a total of $O(m^{\lg_4 3}\gamma^{-2} \ln m) = O(m^{0.8}\gamma^{-2})$ times on a training set of $m$ samples. This should be compared to AdaBoost that only invokes the weak learner $O(\gamma^{-2} \ln m)$ times to achieve the sample complexity stated in (1).

An alternative sample optimal weak-to-strong learner was given by Larsen [19] as a corollary of a proof that Bagging [5] is an optimal PAC learner in the realizable setting. Concretely, his work gives a weak-to-strong learner with an optimal sample complexity while only invoking the weak-learner a total of $O(\gamma^{-2} \ln(m/\delta) \ln m)$ times on a training set of $m$ samples.

A natural question is whether the sample complexity of AdaBoost shown in (1) can be improved to match the optimal sample complexity by a better analysis. Since AdaBoost only invokes its weak learner $O(\gamma^{-2} \ln m)$ times on $m$ samples, this would be an even more efficient optimal weak-to-strong learner. Unfortunately, work by Høgsgaard et al. [15] shows that AdaBoost's sample complexity is sub-optimal by at least a $\ln(1/\varepsilon)$ factor, i.e.

$$m_{\mathrm{Ada}}(\varepsilon, \delta) = \Omega\left(\frac{d \ln(1/\varepsilon)}{\gamma^2 \varepsilon} + \frac{\ln(1/\delta)}{\varepsilon}\right). \tag{2}$$

It thus remains an intriguing task to design weak-to-strong learners that have an optimal sample complexity and yet match the runtime guarantees of AdaBoost. Furthermore, do the theoretical improvements translate to practice? Or are there large hidden constant factors in the $O(\cdot)$-notation? And how does it vary among the different weak-to-strong learners?

## 1.1 Our Contributions

In this work, we first present a new weak-to-strong learner with an optimal sample complexity (at least in expectation). The algorithm, called MAJORITY-OF-5 and shown as Algorithm 1, is extremely simple: Partition the training set into 5 disjoint pieces of size $m/5$ and run AdaBoost on each to produce voting classifiers $f_1, \ldots, f_5$. Finally, combine them by taking a majority vote $g(x) = \text{sign}(\sum_{t=1}^{5} f_t(x))$. This simple algorithm only invokes the weak learner $O(\gamma^{-2} \ln m)$ times, asymptotically matching AdaBoost and improving over previous optimal weak-to-strong learners. Furthermore, since each invocation of AdaBoost is on a training set of only $m/5$ samples, it is at least as fast as AdaBoost, even when considering constant factors. It is even trivial to parallelize the algorithm among up to 5 machines/threads.

---

**Algorithm 1:** MAJORITY-OF-5$(S, \mathcal{W})$

**Input:** Training set $S = (x_1, y_1), \ldots, (x_m, y_m)$. Weak learner $\mathcal{W}$.
**Result:** Hypothesis $g : \mathcal{X} \to \{-1, 1\}$.
1 Partition $S$ into 5 disjoint pieces $S_1, \ldots, S_5$ of size $m/5$.
2 **for** $t = 1, \ldots, 5$ **do**
3 $\quad$ Run AdaBoost on $S_t$ with $\mathcal{W}$ to obtain $f_t : \mathcal{X} \to \{-1, 1\}$.
4 $g \leftarrow \text{sign}(\sum_t f_t)$.
5 **return** $g$

---

The concrete guarantees we give for MAJORITY-OF-5 are as follows

**Theorem 1.** *For any distribution $\mathcal{D}$ over $\mathcal{X} \times \{-1, 1\}$ and any $\gamma$-weak learner $\mathcal{W}$ using a hypothesis set $\mathcal{H}$ of VC-dimension $d$, it holds for a training set $S \sim \mathcal{D}^m$ that running MAJORITY-OF-5 on $S$ to obtain a hypothesis $g$ satisfies*

$$\mathbb{E}[\text{er}_{\mathcal{D}}(g)] = O\left(\frac{d}{\gamma^2 m}\right).$$

In particular, Theorem 1 implies that $\mathbb{E}[\text{er}_{\mathcal{D}}(g)] \leq \varepsilon$ when given $m = \Theta(d/(\gamma^2 \varepsilon))$ samples. This is a slightly weaker guarantee than the alternative optimal weak-to-strong learners in the sense that we do not provide high probability guarantees (i.e. with probability $1 - \delta$). On the other hand, our algorithm is extremely simple and has a running time comparable to AdaBoost. Furthermore, the proof that AdaBoost is sub-optimal (see (2)) shows that even the expected error of AdaBoost is sub-optimal by a logarithmic factor. It is interesting that combining a constant number of voting classifiers trained by AdaBoost makes it optimal when a single AdaBoost is provably sub-optimal. Let us also comment that the analysis of Algorithm 1 is based on recent work by Aden-Ali et al. [1] on optimal PAC learning in the realizable setting, demonstrating new applications of their techniques. Furthermore, we believe the number 5 is an artifact of our proof and we conjecture that it can be replaced with 3 by giving a better generalization bound for *large margin* voting classifiers. See Section 3 for further details.

**Empirical Comparison.** Our second contribution is a pilot empirical study, which gives the first empirical comparison between the alternative optimal weak-to-strong learners, both the algorithm of Larsen and Ritzert [20], the Bagging+Boosting based algorithm [19], our new MAJORITY-OF-5 algorithm, as well as classic AdaBoost. We give the full details of the alternative algorithms in Section 2. In our experiments, we compare their performance both on real-life data as well as the data distribution used by Høgsgaard et al. [15] in their proof that AdaBoost is sub-optimal as shown in (2). Our pilot empirical study give an indication that our new algorithm MAJORITY-OF-5 may outperform previous algorithms on large data sets, whereas Bagging+Boosting performs best on small data sets. See Section 4 for further details and the results of these experiments.

## 2 Previous Optimal Weak-to-Strong Learners

In this section, we present the two previous optimal weak-to-strong learners. The first such algorithm, by Larsen and Ritzert [21], builds on a sub-sampling technique due to Hanneke [14] in his seminal work on optimal PAC learning in the realizable setting. This sub-sampling technique, named SUBSAMPLE, is shown as Algorithm 2.

---

**Algorithm 2:** SUBSAMPLE$(S, T)$

**Input:** Training set $S$, Stash $T$
**Result:** List of training sets $L$.
**1 if** $|S| < 4$ **then**
**2** $\quad$ Let $L$ contain the single training set $S \cup T$.
**3** $\quad$ **return** $L$
**4** Partition $S$ into $4$ disjoint pieces $S_0, S_1, S_2, S_3$ of size $|S|/4$ each.
**5** Let $L$ be an empty list.
**6** Append SUBSAMPLE$(S_0, T \cup S_2 \cup S_3)$ to $L$.
**7** Append SUBSAMPLE$(S_0, T \cup S_1 \cup S_3)$ to $L$.
**8** Append SUBSAMPLE$(S_0, T \cup S_1 \cup S_2)$ to $L$.
**9 return** $L$

---

Given a training set $S$, SUBSAMPLE generates a list $L$ of subsets $S_i \subset S$. The list $L$ has size $m^{\lg_4 3} \approx m^{0.79}$ when invoking SUBSAMPLE$(S, \emptyset)$ for a training set $S$ of size $m$. Larsen and Ritzert now give an optimal weak-to-strong learner using SUBSAMPLE as a sub-routine as shown in Algorithm 3.

---

**Algorithm 3:** LARSENRITZERT$(S, \mathcal{W})$

**Input:** Training set $S = (x_1, y_1), \ldots, (x_m, y_m)$. Weak learner $\mathcal{W}$.
**Result:** Hypothesis $g : \mathcal{X} \to \{-1, 1\}$.
**1** Invoke SUBSAMPLE$(S, \emptyset)$ to obtain list $L = S_1, \ldots, S_k$.
**2 for** $t = 1, \ldots, k$ **do**
**3** $\quad$ Run AdaBoost on $S_t$ with $\mathcal{W}$ to obtain $f_t : \mathcal{X} \to \{-1, 1\}$.
**4** $g \leftarrow \text{sign}(\sum_t f_t)$.
**5 return** $g$

---

Finally, the algorithm by Larsen [19] based on Bagging (a.k.a. Bootstrap Aggregation) by Breiman [5], combines AdaBoost with sampling subsets of the training data with replacement. Unlike the algorithm above by Larsen and Ritzert, it requires a target failure probability $\delta$ as input. The algorithm is shown as Algorithm 4.

---

**Algorithm 4:** BAGGEDADABOOST$(S, \mathcal{W}, \delta)$

**Input:** Training set $S = (x_1, y_1), \ldots, (x_m, y_m)$. Weak learner $\mathcal{W}$. Failure probability $0 < \delta < 1$.
**Result:** Hypothesis $g : \mathcal{X} \to \{-1, 1\}$.
**1 for** $t = 1, \ldots, O(\ln(m/\delta))$ **do**
**2** $\quad$ Let $S_t$ be a set of $m$ independent samples with replacement from $S$.
**3** $\quad$ Run AdaBoost on $S_t$ with $\mathcal{W}$ to obtain $f_t : \mathcal{X} \to \{-1, 1\}$.
**4** $g \leftarrow \text{sign}(\sum_t f_t)$.
**5 return** $g$

---

## 3 Analysis of MAJORITY-OF-5

In this section, we give the proof of Theorem 1, showing that our new algorithm MAJORITY-OF-5 has an optimal expected error. Before giving the formal details, we present the main ideas in our proof. Our analysis is at a high level inspired by recent work of Aden-Ali et al. [1] for

realizable PAC learning. The first ingredient we need is the notion of margins. For a voting classifier $f(x) = \text{sign}(\sum_{h \in \mathcal{H}} \alpha_h h(x))$ with $\alpha_h \geq 0$ for all $h$, consider the function $f'(x) = \sum_{h \in \mathcal{H}} \alpha'_h h(x)$ with $\alpha'_h = \alpha_h / \sum_h \alpha_h$. That is, $f'$ is simply the voting classifier $f$ without the $\text{sign}(\cdot)$ and normalized to have coefficients summing to 1. The margin of $f$ on a sample $(x, c(x))$ is then $c(x)f'(x) \in [-1, 1]$. The margin is 1 if all hypotheses combined by $f$ agree and are correct. It is 0 if half of the mass is on hypothesis that are correct and half of the mass is on the hypothesis that are wrong. We can thus think of the margin as a confidence of the voting classifier. Margins have been extensively studied in the context of boosting and were originally introduced to give theoretical justification for the impressive practical performance of AdaBoost [2]. In particular, there are strong generalization bounds for voting classifiers with large margins [2, 6, 11]. Indeed, the best known sample complexity bound for AdaBoost, as stated in (1), is derived by showing that AdaBoost produces a voting classifier with margins $\Omega(\gamma)$.

Returning to our outline of the proof of Theorem 1, recall that the optimal error for weak-to-strong learning as a function of the number of samples $m$ is $O(d/(\gamma^2 m))$. Now assume we can prove that for a set of $m$ i.i.d. samples from a distribution $\mathcal{D}$, the expected maximum error under $\mathcal{D}$ of any voting classifier that has margins $\Omega(\gamma)$ on all the samples, is no more than $O(\sqrt{d/(\gamma^2 m)})$. This fact follows from previous work on Rademacher complexity. Note that this is sub-optimal compared to our target error by a polynomial factor since $\sqrt{x} \geq x$ for $x$ between 0 and 1. We want to argue that combining 5 instantiations of AdaBoost on disjoint training sets reduces this expected error to optimal $O(d/(\gamma^2 m))$.

For this argument, consider running AdaBoost on $n = m/5$ samples. For any $x \in \mathcal{X}$, consider the probability $p_x = \Pr_{S \sim \mathcal{D}^n}[f_S(x) \neq c(x)]$ where $f_S$ is the hypothesis produced by AdaBoost on $S$ and $c(x)$ is the correct label of $x$. Inspired by Aden-Ali et al. [1], we now partition the input domain $\mathcal{X}$ into sets $R_i$, such that $R_i$ contains all $x$ for which $p_x \in (2^{-i}, 2^{-i+1}]$. The crucial observation is that if we consider $k$ independently trained AdaBoosts, then the probability they all err on $x$ is precisely $p_x^k$. Since a majority vote among 5 classifiers only fails when at least 3 of the involved classifiers fail, combining 5 AdaBoosts intuitively reduces the contribution to the expected error from points $x \in R_i$ to $\Pr_{X \sim \mathcal{D}}[X \in R_i]2^{-3i}$. What remains is thus to argue that $\Pr[X \in R_i]$ is small.

This last step is done by considering the distribution $\mathcal{D}_i$, which is $\mathcal{D}$ conditioned on receiving a sample from $R_i$. The expected number of samples we see from $R_i$ is $m_i = \Pr[X \in R_i]m$. Furthermore, since AdaBoost obtains margins $\Omega(\gamma)$ on all its training data, it in particular obtains margins $\Omega(\gamma)$ on all its samples from $\mathcal{D}_i$. This leads to an error probability of $p_i = O(\sqrt{d/(\gamma^2 m_i)})$ under $\mathcal{D}_i$. But the definition of $R_i$ implies $p_i \geq 2^{-i}$. Hence $\sqrt{d/(\gamma^2 m_i)} = \Omega(2^{-i}) \Rightarrow m_i = O(2^{2i}d/\gamma^2) \Rightarrow \Pr[X \in R_i] = O(2^{2i}d/(\gamma^2 m))$. By summing over all $R_i$, the final expected error is hence

$$\sum_{i=1}^{\infty} \frac{2^{2i}d2^{-3i}}{\gamma^2 m} = O(d/(\gamma^2 m)).$$

This completes the proof overview. Let us end by making a few remarks. First, it is worth noting that the generalization bound $O(\sqrt{d/(\gamma^2 m)})$ seems much worse than the bound in (1) claimed for AdaBoost and other voting classifiers with large margins. Unfortunately, if we examine (1) carefully and state $\varepsilon$ as a function of $m$, we get $\varepsilon = O(d \ln(m/d) \ln m/(\gamma^2 m))$. The problem is that the two log-factors are not bounded by a polynomial in $d/(\gamma^2 m)$. In particular for $m = Cd/\gamma^2$ with $C > 0$ a constant, any polynomial in $d/(\gamma^2 m)$ must be constant. But $\ln m = \ln(Cd/\gamma^2)$ is not a constant independent of $d$ and $\gamma$. Thus we have to use the generalization bound with $\sqrt{\cdot}$ that fortunately is within a polynomial factor of optimal for the full range of $m$. Let us also comment that if the lower bound for AdaBoost stated in (2) is tight also for $\gamma$-margin voting classifiers, i.e. matched by an upper bound, then it suffices to take a majority of 3 AdaBoosts for optimal sample complexity.

This concludes the description of the high level ideas in our proof.

## 3.1 Formal Analysis

We now give the formal details of the proof. We start by introducing some notation.

**Preliminaries.** For a hypothesis set $\mathcal{H}$, we let $\Delta(\mathcal{H})$ denote the set of *linear* classifiers using hypothesis from $\mathcal{H}$ that is

$$\Delta(\mathcal{H}) = \left\{ f \in \{-1,1\}^{\mathcal{X}} : f = \sum_{h \in \mathcal{H}} \alpha_h^f h, \forall h \in \mathcal{H}, \alpha_h^f \geq 0, \sum_{h \in \mathcal{H}} \alpha_h^f = 1 \right\}.$$

Note that we have termed these *linear* classifiers rather than voting classifier, to distinguish that we have not yet applied a $\text{sign}(\cdot)$ and insist on normalizing the coefficients so they sum to 1. We define the $\text{sign}$ function as being 1 when the value is non-negative (so also 1 when $x = 0$) and $-1$ when negative.

We let $c \in \{-1,1\}^{\mathcal{X}}$ be a true labeling that we are trying to learn. For a distribution $\mathcal{D}$ over $\mathcal{X}$, we define the expected loss of $f \in [-1,1]^{\mathcal{X}}$ as $\text{er}_{\mathcal{D}}(f) := \mathbb{E}_{X \sim \mathcal{D}}[1\{\text{sign}(f)(X) \neq c(X)\}]$ and we will use $S \in \mathcal{X}^m$ to denote a point set of $m$ i.i.d. samples from $\mathcal{D}$ i.e. $S \sim \mathcal{D}^m$ (we see the point set as a vector so we allow repetition of points). Define for any $k \in \mathbb{N}$ the majority of $k$ linear classifiers $f_1, \ldots, f_k \in [-1,1]^{\mathcal{X}}$ as $Maj(f_1, \ldots, f_k)(x) = \text{sign}(\sum_i \text{sign}(f_i(x)))$. Let $s$ be a set of points, i.e. $s \in \cup_{i=1}^{\infty} \mathcal{X}^i$ and $c(s) \in \{-1,1\}^{|s|}$ the labeling of the points that $s$ contains with $c$, that is $c(s)_i = c(s_i)$. We define a $\gamma$-margin classifier algorithm $f : \cup_{i=1}^{\infty}(\mathcal{X} \times \{-1,1\})^i \Rightarrow [-1,1]^{\mathcal{X}}$ to be a mapping that takes as input a point set with labels $(s, c(s)) \in \cup_{i=1}^{\infty}(\mathcal{X} \times \{-1,1\})^i$ and outputs a function $f(s, c(s))(\cdot)$ from $\mathcal{X}$ to the interval $[-1,1]$, where $f(s, c(s))(\cdot)$ is such that for $x \in s$, $f(s, c(s))(x)c(x) \geq \gamma$. In the following we will use $f_s$ to denote $f(s, c(s))$ and write $f \in \Delta(\mathcal{H})$ if for any $(s, c(s)) \in \cup_{i=1}^{\infty}(\mathcal{X} \times \{-1,1\})^i$ we have that the output of the $\gamma$-margin classifier algorithm $f(s, c(s))$ is in $\Delta(\mathcal{H})$.

**Analysis.** We now prove Theorem 1, which is a direct consequence of the following Corollary 2. Corollary 2 states that running a $\gamma$-margin classifier algorithm on 5 disjoint training sets of size $m$ and forming the majority vote of the produced 5 classifiers, has the optimal $O(d/(\gamma^2 m))$ expected error. Since AdaBoost, after $O(\ln(m)/\gamma^2)$ iterations, has $\Omega(\gamma)$ margins on all points [26] [Section 5.4.1], Corollary 2 gives the claim in Theorem 1. Alternatively one could run AdaBoost$_v^*$ [24] instead of AdaBoost. The proof of Corollary 2 follows the method used in [1] where the authors show a similar bound for PAC learning in the realizable setting. We now state Corollary 2.

**Corollary 2.** *For any distribution $\mathcal{D}$ over $\mathcal{X}$, hypothesis set $\mathcal{H}$ with VC-dimension $d$, i.i.d. point sets $S_1, \ldots, S_5$ from $\mathcal{D}^m$, margin $0 < \gamma \leq 1$ and $f \in \Delta(\mathcal{H})$ being a $\gamma$-margin classifier algorithm, we have that*

$$\mathbb{E}_{S_1, \ldots, S_5 \sim \mathcal{D}^m}[\text{er}_{\mathcal{D}}(Maj(f_{S_1}, \ldots, f_{S_5}))] = O\left(\frac{d}{\gamma^2 m}\right).$$

The result in Corollary 2 is primarily a consequence of the following Lemma 3, which says that, in expectation, the probability that $f_{S_1}, f_{S_2}, f_{S_3}$ all misclassifying a sample from $\mathcal{D}$ is $O(d/(\gamma^2 m))$. We now state Lemma 3 and give the proof of Corollary 2. We postpone the proof of Lemma 3 to later in this section.

**Lemma 3.** *For any distribution $\mathcal{D}$ over $\mathcal{X}$, hypothesis set $\mathcal{H}$ with VC-dimension $d$, i.i.d. point sets $S_1, S_2, S_3$ from $\mathcal{D}^m$, margin $0 < \gamma \leq 1$ and $f \in \Delta(\mathcal{H})$ being a $\gamma$-margin classifier algorithm we have that*

$$\mathbb{E}_{S_1, S_2, S_3 \sim \mathcal{D}^m}\left[\mathbb{P}_{X \sim D}\left[\cap_{i=1}^3 \{\text{sign}(f_{S_i}(X)) \neq c(X)\}\right]\right] = O\left(\frac{d}{\gamma^2 m}\right).$$

*Proof of Corollary 2.* For the majority of $f_{S_1}, \ldots, f_{S_5}$ to fail on an $x \in \mathcal{X}$, it must be the case that at least 3 of the trained $\gamma$-margin classifiers $f_{S_i}$ have $\text{sign}(f_{S_i}(x)) \neq c(x)$. Using this combined with the $S_i$'s being i.i.d. we get that

$$\mathbb{E}_{S_1, \ldots, S_5 \sim \mathcal{D}^m}[\text{er}_{\mathcal{D}}(Maj(f_{S_1}, \ldots, f_{S_5}))]$$

$$\leq \sum_{1 \leq j_1 < j_2 < j_3 \leq 5} \mathbb{E}_{S_{j_1}, S_{j_2}, S_{j_3} \sim \mathcal{D}^m}\left[\mathbb{P}_{X \sim D}\left[\cap_{i=1}^3 \{\text{sign}(f_{S_{j_i}}(X)) \neq c(X)\}\right]\right]$$

$$\leq \mathbb{E}_{S_1, S_2, S_3 \sim \mathcal{D}^m}\left[\mathbb{P}_{X \sim D}\left[\cap_{i=1}^3 \{\text{sign}(f_{S_i}(X)) \neq c(X)\}\right]\right] \sum_{1 \leq j_1 < j_2 < j_3 \leq 5} 1.$$

Now using Lemma 3 and $\binom{5}{3} = O(1)$, we get that the above is $O\left(\frac{d}{\gamma^2 m}\right)$ as claimed. $\quad\square$

We now move on to prove Lemma 3. For this we need Lemma 4 which we now state and give the proof in Appendix A.

**Lemma 4.** *Let $a > 1$ denote a universal constant. For $\mathcal{D}$ a distribution, $R$ a subset of $\mathcal{X}$ such that $\mathbb{P}[R] := \mathbb{P}_{X \sim \mathcal{D}}[X \in R] \neq 0$, hypothesis set $\mathcal{H}$ with VC-dimension $d$, $S$ a point set of $m$ i.i.d. points from $\mathcal{D}$, margin $0 < \gamma \leq 1$ and $f \in \Delta(\mathcal{H})$ being a $\gamma$-margin classifier algorithm we have that*

$$\mathbb{E}_S\left[\mathbb{E}_{X \sim \mathcal{D}_R}[1\{f_S(X) \neq c(X)\}]\right] = \mathbb{E}_S\left[\mathrm{er}_{\mathcal{D}_R}(f_S)\right] \leq \sqrt{\frac{ad}{\mathbb{P}[R]\gamma^2 m}}.$$

*where we use $\mathcal{D}_R$ to denote the conditional distribution on the subset $R$. That is, for any measurable function $g$, $\mathbb{E}_{X \sim \mathcal{D}_R}[g(X)] := \mathbb{E}_{X \sim \mathcal{D}}[g(X)1\{X \in R\}]/\mathbb{P}_{X \sim \mathcal{D}}[X \in R]$.*

*Proof of Lemma 3.* For $x \in \mathcal{X}$ let $p_x = \mathbb{E}_{S \sim \mathcal{D}^m}[1\{f_S(x) \neq c(x)\}]$ and define for $i = 1, \ldots$ the sets $R_i = \{x \in \mathcal{X} : p_x \in (2^{-i}, 2^{-i+1}]\}$. Now using Tonelli, and that $S_1, S_2, S_3$ are i.i.d. with distribution $\mathcal{D}^m$, and that $p_x \leq 2^{-i+1}$ for $x \in R_i$ we get that

$$\mathbb{E}_{S_1,\ldots,S_3 \sim \mathcal{D}^m}\left[\mathbb{P}_{X \sim \mathcal{D}}\left[\cap_{i=1}^3\{f_{S_i}(X) \neq c(X)\}\right]\right]$$

$$=\mathbb{E}_{X \sim \mathcal{D}}\left[\mathbb{E}_{S_1,\ldots,S_3 \sim \mathcal{D}^m}\left[1\{\cap_{i=1}^3\{f_{S_i}(X) \neq c(X)\}\}\right]\right]$$

$$=\mathbb{E}_{X \sim \mathcal{D}}\left[p_X^3\right] = \sum_{i=1}^{\infty} \mathbb{E}_{X \sim \mathcal{D}}\left[p_X^3 \mid X \in R_i\right]\mathbb{P}[X \in R_i] \leq 2^3 \sum_{i=1}^{\infty} 2^{-3i}\mathbb{P}[X \in R_i],$$

thus if we can show that there exists a universal constant $c' > 0$ such that $\mathbb{P}_{X \sim \mathcal{D}}[X \in R_i] \leq \frac{c'd2^{2i}}{\gamma^2 m}$ we get that

$$\mathbb{E}_{S_1,\ldots,S_3}\left[\mathbb{P}_{X \sim \mathcal{D}}\left[\cap_{i=1}^3\{f_{S_i}(X) \neq c(X)\}\right]\right] \leq 2^3 \frac{c'd}{\gamma^2 m} \sum_{i=1}^{\infty} 2^{-i} = O\left(\frac{d}{\gamma^2 m}\right),$$

and we are done. Thus assume for contradiction that $\mathbb{P}_{X \sim \mathcal{D}}[X \in R_i] > \frac{c'd2^{2i}}{\gamma^2 m}$, for $c' > 1$ to be chosen large enough. Using Lemma 4 we have that there exist a universal constant $a > 1$ such that

$$\mathbb{E}_{S \sim \mathcal{D}^m}\left[\mathrm{er}_{\mathcal{D}_{R_i}}(f_S)\right] \leq \sqrt{\frac{ad}{\mathbb{P}[R_i]\gamma^2 m}}.$$

By Tonelli, the definition of $p_x$ and that for $x \in R_i$ we have $p_x \in (2^{-i}, 2^{-i+1}]$ we get that

$$\mathbb{E}_{S \sim \mathcal{D}^m}\left[\mathrm{er}_{\mathcal{D}_{R_i}}(f_S)\right] = \mathbb{E}_{X \sim \mathcal{D}_{R_i}}\left[\mathbb{E}_{S \sim \mathcal{D}^m}[1\{f_S(X) \neq c(X)\}]\right] = \mathbb{E}_{X \sim \mathcal{D}_{R_i}}[p_X] \geq 2^{-i}.$$

Combining the above lower and upper bound on $\mathbb{E}_{S \sim \mathcal{D}^m}\left[\mathrm{er}_{\mathcal{D}_{R_i}}\right]$ and $\mathbb{P}_{X \sim \mathcal{D}}[X \in R_i] > \frac{c'd2^{2i}}{\gamma^2 m}$ we get that

$$1 \leq 2^i\sqrt{\frac{ad}{\mathbb{P}[R_i]\gamma^2 m}} \leq 2^i\sqrt{\frac{ad}{\frac{c'd2^{2i}}{\gamma^2 m}\gamma^2 m}} \leq \sqrt{\frac{a}{c'}},$$

which for $c'$ sufficiently large is strictly less than 1, thus we reached a contradiction. Hence it must be the case that $\mathbb{P}_{X \sim \mathcal{D}}[X \in R_i] \leq \frac{c'd2^{2i}}{\gamma^2 m}$, which concludes the proof of Lemma 3. $\qquad\square$

## 4 Experiments

In this section, we present the results of our pilot empirical study between the different sample optimal weak-to-strong learners. We compare the algorithms on five different data sets. The first four are standard binary classification data sets and are the same data sets used in [12], whereas the last is a synthetic binary classification data set developed from the lower bound [15] showing that AdaBoost is sub-optimal. For all real world data sets, we have shuffled the samples and randomly set aside 20% to use as test set. The weak learner we use for these is the scikit-learn `DecisionTreeClassifier` with `max_depth=1`. This is default for the implementation of AdaBoost in scikit-learn, which is the implementation used in our experiments. We describe the data sets in greater detail below.

- **Higgs** [29]: This data set represents measurements from particle detectors, and the labels tells whether they come from a process producing Higgs bosons or if they were a background process. The data set consists of 11 million labeled samples. However, we focus on the first 300,000 samples. Each sample consists of 28 features, where 7 of these are derived from the other 21.

- **Boone** [23]: In this data set, we try to distinguish electron neutrinos from muon neutrinos. The data set consists of 130,065 labeled samples. Each sample consists of 50 features.

- **Forest Cover** [4]: In this data set, we try to determine the forest cover type of 30 x 30 meter cells. The data set actually has 7 different forest cover types, so we have removed all samples of the 5 most uncommon to make it into a binary classification problem. This leaves us with 495,141 samples. Each sample consists of 54 features such as elevation, soil-type and more.

- **Diabetes** [28]: In this data set, we try to determine whether a patient has diabetes or not from features such as BMI, insulin level, age and so on. This is the smallest real-world data set, consisting of only 768 samples. Each sample consists of 8 features.

- **Adversarial** [15]: This data set, as well as the weak learner, have been developed using the lower bound instance in [15]. Concretely, the data set consists of 1024 uniform random samples from the universe $\mathcal{X} = \{1, \ldots, 350\}$. Every element of the input domain has the label 1, but all weak-to-strong learners are run simply by giving them access to a weak learner. The weak learner is adversarially designed. When it is queried with a weighing of the training data set, it computes the set $T$ containing the first 20 points from the input domain that receive zero mass under the query weighing. It then searches through a set of hypotheses (chosen randomly) and returns the hypothesis with the worst performance on $T$, while respecting that it must have error no more than $1/2 - \gamma$ under the query weighing and having at least $1/2 + \gamma$ error on $T$. Finally, the hypothesis set contains an additional special hypothesis $h_0$ that is correct (returns 1) on all but the last 20 points of the input domain. This hypothesis is used to handle queries where none of the randomly chosen hypotheses have advantage $\gamma$. We refer the reader to [15] for further details and intuition on why this construction is hard for AdaBoost.

The data sets represent both large and small training sets. For each data set, we run simple AD-ABOOST (accuracy shown as a blue horizontal line in the plots), BAGGEDADABOOST (Algorithm 4), LARSENRITZERT (Algorithm 3), and our new MAJORITY-OF-X (for X varying from 3 to 29). In our experiments, we vary the number of AdaBoosts trained by each weak-to-strong learner from 3 to 29, instead of merely following the theoretical suggestions. Each of these voting classifiers is then trained for 300 rounds on its respective input. This has been repeated 5 times with different random seeds, so the plots indicate the average accuracy across these 5 runs. For Algorithm 3 that creates $m^{\lg_4 3}$ sub-samples, we use the full set of sub-samples on the two small data sets Diabetes and Adversarial. For the three large data sets Higgs, Boone and Forest Cover, this creates a huge overhead in running time and we instead randomly sample without replacement from among the sub-samples resulting from the SUBSAMPLE procedure (Algorithm 2). This is the reason for the non-constant behavior of this algorithm in the corresponding experiments. For BAGGEDADABOOST, we have chosen to sample 95% of the samples (with replacement) in our experiments. The results of the experiments on the three large data sets are shown in Fig. 1.

The results in Fig. 1 gives an initial suggestion that our new algorithm with disjoint training sets might have an advantage on large data sets. Quite surprisingly, we see that the two other optimal weak-to-strong learners perform no better, or even worse, than standard AdaBoost. Experiments on the small Diabetes data set, as well as the Adversarially designed data set, are shown in Fig. 2.

The results in Fig. 2 suggest that our new algorithm may perform poorly on small training sets. This makes sense, as the training data for each weak learner is extremely small on these data sets. Instead, we find that the Bagging based variant outperforms classic AdaBoost. Since Bagging has a relative small overhead compared to simple AdaBoost, this suggests running both our new algorithm MAJORITY-OF-X and BAGGEDADABOOST and using a validation set to pick the best classifier. We hope these first experiments may inspire future and more extensive empirical comparisons between the various weak to strong learners.

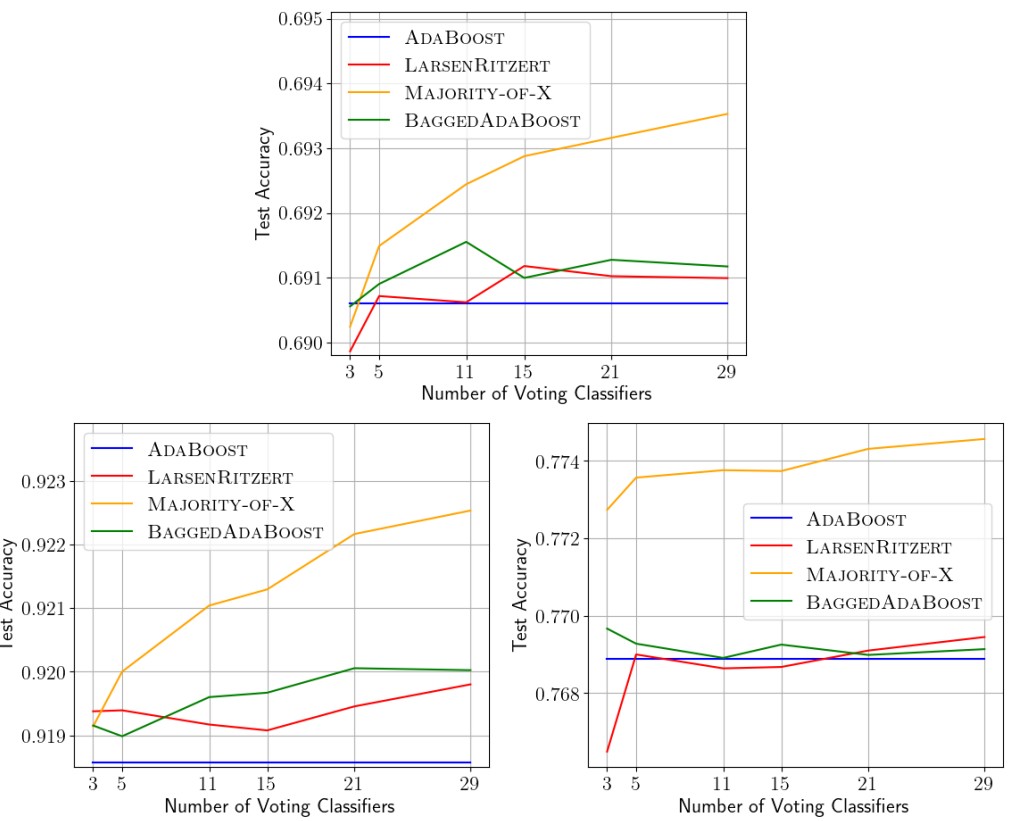

Figure 1: Top is Higgs, Left plot is Boone. Right plot is Forest Cover

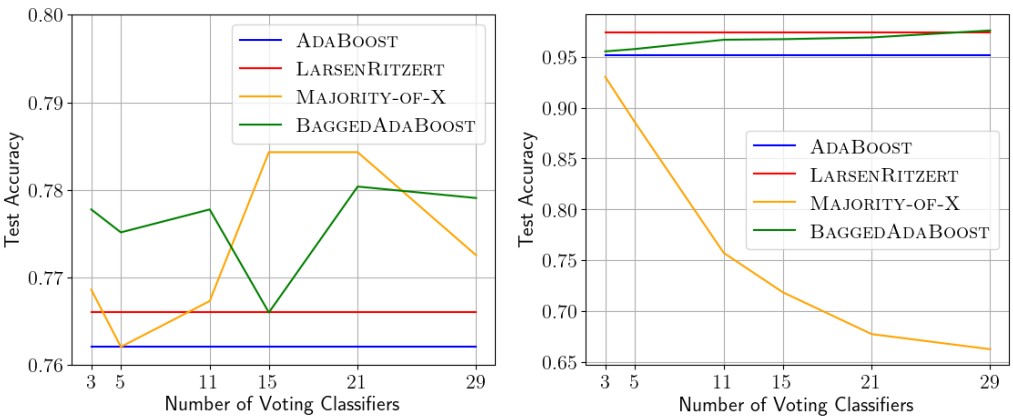

Figure 2: Left plot is Diabetes. Right plot is Adversarial

## 5 Limitations

Our main results are proved under the theoretical assumption of i.i.d. training samples as well as access to a weak-learner that always obtains an advantage of $\gamma$ over random guessing. Since these might not be realistic assumptions, we also performed an empirical evaluation of our algorithm. Due to computational constraints, we have only been able to run experiments on 5 data sets. We have thus been careful not to over-emphasize the practical implications of our results. In all circumstances, we view the theoretical contributions as the main novelty of this work.

## Acknowledgment

This research is co-funded by the European Union (ERC, TUCLA, 101125203) and Independent Research Fund Denmark (DFF) Sapere Aude Research Leader Grant No. 9064-00068B. Views and opinions expressed are however those of the author(s) only and do not necessarily reflect those of the European Union or the European Research Council. Neither the European Union nor the granting authority can be held responsible for them.

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

# A  Proof of Lemma 4

We now give the proof of Lemma 4. For this we need the following Corollary 5 that gives a high probability bound on the error for all classifiers in $\Delta(\mathcal{H})$ that have $\gamma$ margins on a training set $S$.

**Corollary 5.** *Let $C > 1$ denote a universal constant. For hypothesis set $\mathcal{H}$ with VC-dimension $d$, distribution $\mathcal{D}$, margin $0 < \gamma \leq 1$, failure probability $0 < \delta < 1$, and a point set $S \sim \mathcal{D}^m$, we have with probability at least $1 - \delta$ over $S$, that any $f \in \Delta(\mathcal{H})$ such that $f(x)c(x) \geq \gamma$ for all $x \in S$ satisfies*

$$\mathrm{er}_{\mathcal{D}}(f) \leq \sqrt{\frac{2Cd}{\gamma^2 m}} + \sqrt{\frac{2\ln(2/\delta)}{m}}.$$

Corollary 5 follows for instance by a modification of [26] [page 107-111] due to [3] and using the stronger bound on the Rademacher Complexity of $O(\sqrt{d/m})$ due to [8] [See e.g. theorem 7.2 [13] ]. With Corollary 5 in place we now give the proof of Lemma 4.

*Proof of Lemma 4.* If $\mathbb{P}[R]\, m \leq d/\gamma^2$ we have that

$$\sqrt{\frac{d}{\gamma^2 \mathbb{P}[R]\, m}} \geq 1,$$

and the claim holds since $\mathrm{er}_{\mathcal{D}_R}(f_S)$ is always less than 1. Thus we assume from now on that $\mathbb{P}[R]\, m > d/\gamma^2$. We now define the events $E_i := \{|\{S \cap R\}| = i\}$ for $m \geq i \geq \mathbb{P}[R]\, m/2$ and $E = \cup_{i \geq \mathbb{P}[R] m/2} E_i = \{|S \cap R| \geq \mathbb{P}[R]\, m/2\}$. We further define $X_j = 1\{S_j \in R\}$ for $j = 1, \ldots, m$ and notice that these are i.i.d. $\{0,1\}$-random variables where $X = \sum_j X_j$ has expectation $\mathbb{P}[R]\, m$ and the event $\sum_j X_j \geq \mathbb{P}[R]\, m/2$ is contained in $E$. Thus by a Chernoff bound and that $\exp(-x) \leq 1/x$ for $x > 0$, we get that

$$\mathbb{P}_S[E] \geq 1 - \exp\left(-\mathbb{P}[R]\, m/8\right) \geq 1 - \frac{8}{\mathbb{P}[R]\, m}.$$

We thus have $\mathbb{P}_S\left[\bar{E}\right] \leq 8/(\mathbb{P}[R]\, m)$. Since we assumed that $\mathbb{P}[R]\, m \geq d/\gamma^2 \geq 1$ and using $0 \leq x \leq \sqrt{x}$ for $x \leq 1$, this further implies that

$$\mathbb{P}_S\left[\bar{E}\right] \leq 8 \cdot \sqrt{\frac{d}{\gamma^2 \mathbb{P}[R]\, m}}.$$

We will soon show that $\mathbb{E}_S\left[\mathrm{er}_{\mathcal{D}_R}(f_S)|E_i\right] \leq 16 \cdot \sqrt{\frac{16Cd}{\gamma^2 \mathbb{P}[R]m}}$ for $m \geq i \geq \mathbb{P}[R]\, m/2$ for a universal constant $C \geq 1$. Now using these two relations combined with the law of total expectation on the partition $\bar{E}, E_{\mathbb{P}[R]m/2}, \ldots, E_m$ and that $\mathrm{er}_{\mathcal{D}_R} \leq 1$, we get that

$$\mathbb{E}_S\left[\mathrm{er}_{\mathcal{D}_R}(f_S)\right] = \sum_{i \geq \mathbb{P}[R]m/2} \mathbb{E}_S\left[\mathrm{er}_{\mathcal{D}_R}(f_S)|E_i\right]\mathbb{P}_S[E_i] + \mathbb{E}_S\left[\mathrm{er}_{\mathcal{D}_R}(f_S)|\bar{E}\right]\mathbb{P}_S\left[\bar{E}\right]$$

$$\leq 16 \cdot \sqrt{\frac{16Cd}{\gamma^2 \mathbb{P}[R]\, m}}\mathbb{P}[E] + 8 \cdot \sqrt{\frac{d}{\gamma^2 \mathbb{P}[R]\, m}}$$

$$\leq 24 \cdot \sqrt{\frac{16Cd}{\gamma^2 \mathbb{P}[R]\, m}},$$

as claimed in Lemma 4 with the universal constant $a = 24^2 \cdot 16C$. Thus we have to show that $\mathbb{E}_S\left[\mathrm{er}_{\mathcal{D}_R}(f_S)|E_i\right] \leq 16 \cdot \sqrt{\frac{16Cd}{\gamma^2 \mathbb{P}[R]m}}$ for $m \geq i \geq \mathbb{P}[R]\, m/2$. So consider such an $i$. Since $\mathrm{er}_{\mathcal{D}_R}(f_S)$ is a non-negative random variable, we have that

$$\mathbb{E}_S\left[\mathrm{er}_{\mathcal{D}_R}(f_S) \mid E_i\right] = \int_0^\infty \mathbb{P}_S\left[\mathrm{er}_{\mathcal{D}_R}(f_S) \geq x \mid E_i\right] dx$$

We will thus upper bound this integral. Now conditioned on $E_i$, we know that $S$ contains $i$ points that are samples according to $\mathcal{D}_R$ and that $f_S$ on these examples has all margins at least $\gamma$. Thus we have by Corollary 5 that with probability at least $1 - \delta$ over $S$, it holds that

$$\mathrm{er}_{\mathcal{D}_R}(f_S) \leq \sqrt{\max\left\{\frac{8Cd}{\gamma^2 i}, \frac{8\ln(2/\delta)}{i}\right\}} \tag{3}$$

where $C \geq 1$ is a universal constant. For ease of notation let $r_i = \sqrt{\frac{8Cd}{\gamma^2 i}}$. We notice that $r_i \leq \sqrt{\frac{16Cd}{\gamma^2 \mathbb{P}[R]m}}$ since $i$ is assumed to be greater than $\mathbb{P}[R]m/2$. Using this, we get that

$$\int_0^\infty \mathbb{P}_S\left[\mathrm{er}_{\mathcal{D}_R}(f_S) \geq x \mid E_i\right] dx \leq r_i + \int_{r_i}^\infty \mathbb{P}_S\left[\mathrm{er}_{\mathcal{D}_R}(f_S) \geq x\right] dx \tag{4}$$

$$\leq \sqrt{\frac{16Cd}{\gamma^2 \mathbb{P}[R]m}} + \int_{r_i}^\infty \mathbb{P}_S\left[\mathrm{er}_{\mathcal{D}_R}(f_S) \geq x\right] dx. \tag{5}$$

Thus if we can show that

$$\int_{r_i}^\infty \mathbb{P}_S\left[\mathrm{er}_{\mathcal{D}_R}(f_S) \geq x\right] dx \leq 15 \cdot \sqrt{\frac{d}{\gamma^2 \mathbb{P}[R]m}},$$

we get by combining this with Eq. (4) that

$$\mathbb{E}_S\left[\mathrm{er}_{\mathcal{D}_R}(f_S) \mid E_i\right] \leq 16 \cdot \sqrt{\frac{16Cd}{\gamma^2 \mathbb{P}[R]m}}$$

which would conclude the proof. Thus we now show $\int_{r_i}^\infty \mathbb{P}_S\left[\mathrm{er}_{\mathcal{D}_R}(f_S) \geq x\right] dx \leq 15 \cdot \sqrt{\frac{d}{\gamma^2 \mathbb{P}[R]m}}$. For this, we do the following non-trivial rewriting of $x$ to make it resemble the second term in the max appearing in (3)

$$x = \sqrt{8\ln\left(\frac{2}{2\exp\left(\frac{-x^2 i}{8}\right)}\right) i^{-1}}.$$

Now for any $x \geq r_i$ we have that

$$\mathbb{P}_S\left[\mathrm{er}_{\mathcal{D}_R}(f_S) \geq x \mid E_i\right] = \mathbb{P}_S\left[\mathrm{er}_{\mathcal{D}_R}(f_S) \geq \max(r_i, x) \mid E_i\right],$$

which combined with the rewriting of $x$ and Eq. (3) with $\delta = 2\exp\left(\frac{-x^2 i}{8}\right)$ and noticing $r_i$ is the first argument of the $\max$ in Eq. (3), we get that

$$\mathbb{P}_S\left[\mathrm{er}_{\mathcal{D}_R}(f_S) \geq x \mid E_i\right] = \mathbb{P}_S\left[\mathrm{er}_{\mathcal{D}_R}(f_S) \geq \max(r_i, x) \mid E_i\right] \leq 2\exp\left(\frac{-x^2 i}{8}\right),$$

for any $x \geq r_i$. Now using the density function of a normal distribution with mean 0 and standard deviation $\sigma$ is equal to $\exp(-\frac{1}{2}(x/\sigma)^2)/(\sigma\sqrt{2\pi})$, and letting $N(0, \sigma)$ denote a normal random variable with mean 0 and standard deviation $\sigma$, we get that

$$\int_{r_i}^\infty \mathbb{P}_S\left[\mathrm{er}_{\mathcal{D}_R}(f_S) \geq x \mid E_i\right] dx \leq \int_{r_i}^\infty 2\exp\left(\frac{-x^2 i}{8}\right) dx$$

$$\leq 2\sqrt{8/(2i)}\sqrt{2\pi} \int_{r_i}^\infty \frac{1}{\sqrt{8/(2i)}\sqrt{2\pi}} \exp\left(-\frac{1}{2}\left(\frac{x}{\sqrt{8/(2i)}}\right)^2\right) dx$$

$$\leq 2\sqrt{8\pi/i} \cdot \mathbb{P}\left[N\left(0, \sqrt{\frac{8}{2i}}\right) \geq r_i\right] \leq 8\sqrt{\pi/(\mathbb{P}[R]m)} \leq 15 \cdot \sqrt{\frac{d}{\gamma^2 \mathbb{P}[R]m}},$$

where the second to last inequality follows from $i \geq \mathbb{P}[R]m/2$ and the last inequality by $\mathbb{P}[R]m \geq d/\gamma^2 \geq 1$. $\qquad\square$

