# OpenReview forum: "The Many Faces of Optimal Weak-to-Strong Learning"
_NeurIPS.cc/2024/Conference — NeurIPS 2024 poster_

### Official Review · Reviewer_p9UC · 2024-06-19

**Soundness:** 3
**Presentation:** 3
**Contribution:** 3
**Rating:** 8
**Confidence:** 4

**Summary:**

This paper presents an efficient and simple weak-to-strong learner that has optimal in-expectation error. In weak-to-strong learning, we are given a dataset of $m$ points from a distribution, and a $\gamma$-weak learner that returns hypotheses from a class of VC dimension $d$. AdaBoost, which is a textbook weak-to-strong learner, makes $O(\ln(m)/\gamma^2)$ total invokations to the weak learner, and the best-known analysis for it shows that it suffers an in-expectation error $O\left(\frac{d\ln(m/d)\ln(m)}{\gamma^2 m}\right)$. Larsen and Ritzert (2022) constructed a weak-to-strong learner, that has expected error $O(d/\gamma^2 m)$. Furthermore, they showed that this is the optimal error that one can obtain from $m$ training examples and a $\gamma$-weak learner. However, the weak-to-strong learner by Larsen and Ritzert (2022) makes $O(m^{0.8}/\gamma^2)$ invokations to the weak learner --- which is exponentially worse than AdaBoost. Another bagging-based-boosting algorithm due to Larsen (2023), which also achieves the optimal expected error of $O(d/\gamma^2m)$, makes only $O((\ln m)^2/\gamma^2)$ invokations to the weak-learner. This is still a log factor worse than AdaBoost. Could we then hope to obtain a tighter analysis of the error of AdaBoost, and show that it obtains the optimal error with only $O(\ln(m)/\gamma^2)$ invokations to the weak learner? Unfortunately, no. Høgsgaard et al. (2023) showed that AdaBoost necessarily suffers an expected error which is at least $\Omega(d\ln(m)/\gamma^2 m)$.

Can we then at least shoot for a different weak-to-strong learner that attains the optimal expected error of $O(d/\gamma^2m)$, and also invokes the weak learner only $O(\ln(m)/\gamma^2)$ many times (which is the AdaBoost gold standard)? This paper answers the question in the affirmative, with a remarkably simple weak-to-strong learner that they call Majority-of-29. The algorithm is exceedingly simple to describe: Partition the training dataset into 29 disjoint sub-samples of size $m/29$ each. Run AdaBoost on each subsample, and return the majority vote over the AdaBoosts. Since each AdaBoost makes only $O(\ln(m)/\gamma^2)$ calls to the weak learner, and we run a constant (29) many AdaBoosts, the total number of calls to the weak learner is $O(\ln(m)/\gamma^2)$ as required. Further, using an analysis similar to the recent majority-of-3-ERMs algorithm of Aden-Ali et al. (2023), the authors are able to show that the expected error of Majority-of-29 is $O(d/\gamma^2m)$. The analysis from that work does not extend in a trivial manner, and the authors are required to make appropriate technical modifications and enhancements. The number 29 emerges from the analysis --- the authors require showing a new generalization bound for margin-based classifiers (they show a generalization bound of the order $O((d/\gamma^2m)^{\alpha})$), for $\alpha=1/14$, and this lets them obtain the result for Majority-of-$g(\alpha)$, where $g(\alpha)=2/\alpha+1$. The authors conjecture that the analysis of the generalization bound could be improved, and a Majority-of-3 might well suffice for optimal error.

Finally, the authors also do a (somewhat-limited) empirical comparison of the the performances of the three optimal weak-to-strong learners mentioned above (LarsenRitzert, Bagging-based-boosting, Majority-of-29) as well as AdaBoost. The authors find that for large datasets, Majority-of-29 outperforms the other optimal weak-to-strong learners. On the smaller datasets, the authors find that Bagging-based-boosting outperforms Majority-of-29.

**Strengths:**

The weak-to-strong learner that the authors propose is optimal, and also requires the fewest calls to the weak learner among all optimal weak-to-strong learners that we know. More importantly, it is exceedingly simple and elegant. It also empirically outperforms the other optimal weak-to-strong learners (at least in the experiments performed by the authors). It is also nice to see that the analysis technique from Aden-Ali et al. (2023) finds new applications. The paper is well-written, sets up the stage (along with relevant prior work) well in the first two sections, and provides a nice high-level summary of the formal analysis in Section 3.

**Weaknesses:**

While the theoretical contribution is substantial and undeniable, arguably, the experimental section is extremely limited (which is okay, and the authors admit this at the end, but this is still a limitation, especially if we want to draw conclusions about the empirical performance of the different weak-to-strong learners). The authors only perform experiments on 4 real-world datasets---there are admittedly many more out there, even just in the UCI repository. Could the authors at least elaborate on their rationale behind choosing the datasets that they did? (e.g., was it a random subset of 4? was it the first 4? was it the best 4 from 20 that they observed this trend on?) How might one believe that there is no cherry-picking of datasets involved? The authors make two conclusions from their experiments: 1) on larger datasets, Majority-of-29 outperforms both Bagging-based-boosting and LarsenRitzert. 2) on smaller datasets, Bagging-based-boosting outperforms Majority-of-29. Importantly, the former conclusion is drawn from results on just 3 datasets, and the latter is drawn from just 1! This can really make one skeptical about whether they should truly believe these conclusions. It is okay that this is just a pilot empirical study, but such claims call for significantly larger empirical validation. Also, please see the questions below.

**Questions:**

1) Do we have reason to believe that $O(\ln(m)/\gamma^2)$ calls to the weak learner is indeed the best gold standard we can hope for? To my understanding, the reason we need $O(\ln(m)/\gamma^2)$ calls to the weak learner in AdaBoost is because we want to use a margin-based generalization bound that expects a classifier to have at least $\Omega(\gamma)$ margin on every training sample---AdaBoost attains this guarantee only after $O(\ln(m)/\gamma^2)$ iterations. But could it perhaps be possible that there is a weak-to-strong learner out there that attains optimal error of $O(d/\gamma^2m)$ with $o(\ln(m)/\gamma^2)$ calls to the weak learner?

2) In the Experiments section, the x-axis in Figures 1 and 2 varies the number X of AdaBoosts trained on disjoint partitions in the Majority-of-X algorithm. But this is not a parameter in the other algorithms (BaggedAdaboost and LarsenRitzert). Hence, I would have expected to see a constant line for these other algorithms in the plots (like how the red and blue lines are constant in Figure 2). Why are there different numbers corresponding to different number of voting classifiers in BaggedAdaboost and LarsenRitzert in Figure 1 (and also for BaggedAdaboost in Figure 2)? Am I missing something?

Minor/Typos: \
Line 133: It is 0 if half of hypotheses are  correct and half are wrong --- this is only true **in a weighted sense** right? \
Line 299: this suggests*

**Limitations:**

The authors have adequately addressed any limitations that I can foresee.

---

> ### Author Rebuttal · Authors · 2024-08-05
>
> We thank you for taking the time to thoroughly assesing the article, asking interesting questions, and suggesting concrete improvements.
>
> Experiments:
> As allotted to in the answer to reviewer KvEC, and as you correctly point out, we should have made it more clear that the experiments are very much a pilot. As you also point out, the main contribution of the paper is the theoretical result.
> Regarding the choice of the 4 real-word-datasets, we simply chose the same as used in "Optimal Minimal Margin Maximization with Boosting" by Gr\o nlund et al. (ICML'19). We did not run on any other data sets and no results were left out, so we wouldn't say that there was any cherry picking done.
>
> Let us also comment on the reason for not conducting more experiments. We are in a theory research group and do not have access to other machines than our laptops. Hence conducting a large number of experiments was simply infeasible. As we also comment on for reviewer KvEC, if you all feel that the paper is stronger as a pure theory paper, we are okay with removing the experiments. In all circumstances, we will further downtime the significance of our experiments.
>
>
> (Question 1) Number of calls to weak learner: This is a very interesting question. Classic work by Freund (Y. Freund. Boosting a weak learning algorithm by majority. Information and Computation, 121(2):256– 285, 1995), and also a recent line of work on parallel boosting (Karbasi and Larsen ALT'24, Luy et al. SODA'24), shows that indeed one needs $\Omega(\gamma^{-2} \log(1/\varepsilon))$ calls to a weak learner to obtain error $\varepsilon$. However, these lower bounds actually require $\exp(d) \geq \gamma^{-1}$. If $d$ is assumed a constant, then work by Alon, Gohen, Hazan and Moran STOC'21 shows that there are algorithms using only $\tilde{O}(\gamma^{-1} \log(1/\varepsilon))$ calls.
>
> (Question 2): Regarding the question about BaggedAdaboost and LarsenRitzert not being one line:
>
> We did attempt to explain this in the paragraph immediately following the description of the data sets. Our apologies if this was not clear enough. Since Bagging is well-defined (but not necessarily optimal) with an arbitrary number of bootstrap sub-samples, we simply run BaggedAdaboost with X many bootstrap samples (random sub-samples of the dataset) for varying X. For LarsenRitzert, as explained in that paragraph, the number of sub-samples needed grows unwieldy for large data sets. Thus we chose to instead sample X of the sub-samples defined by LarsenRitzert without replacement. This was due to computational constraints. Furthermore, we feel the plot is more informative when seeing how more and more combined AdaBoosts improve the accuracy.

---

> > ### Comment · Reviewer_p9UC · 2024-08-07
> > **Response to rebuttal**
> >
> > Thank you for your response. In particular, it is good to know that there was no cherry-picking of datasets involved. I would not argue for removing the experiments section entirely, since there is definitely value in having them; instead, as you say, it would be good to tone down the prose on it.
> >
> > Thanks also for the clarification about $O(\ln m/\gamma^2)$ being the standard, as well as the discrepancy about the axis in the experiment. I would definitely clarify the latter in the prose, since it was quite confusing for me.
> >
> > Finally, it is remarkable that you are able to get the number 29 down to 5. Given this, and the updates that you promise about clarifying and toning down experiments, I am happy to increase my score from 7 -> 8. I maintain that this is a strong contribution and deserves to be accepted. Great work!

---

### Official Review · Reviewer_KvEC · 2024-07-12

**Soundness:** 2
**Presentation:** 1
**Contribution:** 1
**Rating:** 3
**Confidence:** 4

**Summary:**

This paper introduces a new Boosting algorithm, MAJORITY-OF-29, which achieves provably optimal sample complexity and is remarkably simple to implement. The algorithm partitions the training data into 29 disjoint subsets, applies AdaBoost to each subset, and combines the resulting classifiers through a majority vote. This approach not only matches the asymptotic performance of AdaBoost but also improves upon previous weak-to-strong learners in terms of simplicity and runtime efficiency.

**Strengths:**

1. The paper introduces a novel method and provides detailed theoretical analysis.

**Weaknesses:**

2. Existing experiments fail to demonstrate the effectiveness of the proposed method, and there is a lack of analysis and discussion on current experimental results.

**Questions:**

Please see the weakness.

**Limitations:**

Limitations are discussed.

---

> ### Author Rebuttal · Authors · 2024-08-05
>
> Regarding the question about experiments: Let us first re-iterate that our main focus in this work is on the theoretical results, which we believe are strong (the paper is also submitted with a primary area of "Learning Theory").  Perhaps we were not clear enough when claiming that our experimental results should be seen as "indications" (as phrased in Section 1 Empirical Comparison), and thus the experiments should be seen a small complement to our main focus the theoretical bound. We also try to allude to this in section 5 (Limitations). However it is seems like we have not made it clear enough that this is only a pilot empirical study as reviewer p9UC also points out the same things - and that the paper's main contribution is the theoretical result. We thank you for pointing this out and will make it clear in following iterations of the submission.
>
> We were not certain about, if effectiveness in the comment "Existing experiments fail to demonstrate the effectiveness of the proposed method" refers to the reported Test Accuracy or the running time of the algorithms so we have given an answer to both in the following:
>
> Test accuracy: The experiments do not strongly indicate that the Majority-OF-29 (now Majority-OF-5) outperforms other optimal methods LarsenRitzert, BaggedAdaBoost. However, based on the theoretical results, it is also not expected that it outperforms them in terms of accuracy, as they are all asymptotically optimal. The benefit of our new algorithm is that it is simpler, easier to implement and more efficient in terms of computation.
>
> Runtime: For a runtime comparison, we have chosen not to directly include this. One reason for this, is that it will be heavily implementation dependent. In particular, if all invocations of AdaBoost are done in a single thread, then our algorithm will surely be faster in practice, as each data point is only included in precisely one AdaBoost invocation. For the other optimal methods, data points are included in many invocations. However, if properly multi-threaded, it is conceivable that other methods could be made as (or nearly as) efficient as our new algorithm.
>
> Even though we don't feel comfortable reporting the runtimes in the article, the time it takes to fit the given number of AdaBoosts in AdaBoost, Majority-of-X, LarsenRitzert, Bagging can be found in the zip file attached to the article in the folder "results", where for a given dataset and run, the time variable $"fit$_$time"$ describes the time it takes to fit the given number of AdaBoosts in a given a run.

---

### Official Review · Reviewer_ynbn · 2024-07-12

**Soundness:** 4
**Presentation:** 4
**Contribution:** 4
**Rating:** 7
**Confidence:** 5

**Summary:**

The authors present a new boosting algorithm: partition training data into 29 pieces of equal size, run AdaBoost on each, and output the majority vote over them. The authors prove that the sample complexity of MajorityVote29 is optimal and its running time is the same order as AdaBoost. Experimental results are also attached, which corroborate their theoretical findings.

**Strengths:**

- Very strong and interesting result
- Mathematically sound, based by my judgement
- Good presentation, self-contained and well-structured

**Weaknesses:**

N/A

**Questions:**

N/A

---

### Author Rebuttal · Authors · 2024-08-05

We would like to thank all reviewers for their thoughtful reviews.

Let us add one general remark that we will leave to the reviewers whether to include in their evaluation of the submission or not. Recently we found a way to improve the result to the majority of 5 instead of majority of 29. The reason why this improvement is possible is due to us becoming aware of a stronger statement of corollary 5 which improves the previous  $O((d/(\gamma^2m)^{1/14})$ bound to $O((d/(\gamma^2m)^{1/2})$. For the intuition of why this leads to a majority of 5, we recall from the proof sketch that we needed to combine $2/\alpha+1$ many AdaBoosts where $\alpha$ is the number of the exponent that we obtain in corollary 5 ($O((d/(\gamma^2m))^{\alpha}$), which now can be set equal to $\alpha=1/2$ and thus we get that we only need to combine $2/\alpha+1=5$ many AdaBoosts to get the optimal in expectation sample complexity. Fortunately, out experiments also included Majority-of-5 so no significant changes to the paper are needed. The new proof only changes (and simplifies) the material in the appendix.

Also, as we wish to be assessed as a theory paper, so primarily on the theoretical contribution, and added the experiments merely as a very first pilot study, let us add that if the reviewers all feel the paper would be stronger without the experiments, we would like to remove the experiments.

---

### Decision · Program_Chairs · 2024-09-25

**Decision:**

Accept (poster)

**Comment:**

If there is one thing that boosting teaches us, it is the value of simple and original ideas whose use relies on nice technical tricks. This paper is a good example which deserves to be seen by the NeurIPS audience.

Regarding experiments: those have generated substantial discussion, for a paper which is theoretical in nature. Looking at the paper, one can say that there has been substantial work in polishing the theory while the experiments comparatively look a bit "let down" (it only takes a look at Figs 1 and 2) and their presentation could have been substantially simplified. But the merits of the paper are principally in its theory. I concur with the idea that those experiments could be changed. A suggestion: reduce the 1.5 page to 0.5 page that would take place in a "Discussion" section before conclusion, along with the story around Corollary 5 (below) and the limitations of the work, currently a single section. Put all that is not necessary in Appendix (dataset descriptions, etc.), eventually find a summary plot / table, insist on the fact that the experiments are preliminary / pilot, tone down the narrative. Again, the theory is the message.

Regarding the replacement of Corollary 5, there are two strategies: the first would be to just replace Corollary 5 with its stronger version and change the rest. I do not support this because the change has not been reviewed. Another strategy would be to use the "Discussion" Section (above) to add a subsection on a potential change to Corollary 5, with a new Corollary properly stated and how it changed the 29 -> 5 split. Hopefully both Corollaries have the same backbone. Extensive proof in Appendix required and proofchecked.

With this, make the why "The answer is 29" clear from Section 1.1: write that 29 is an integer that allows for the appropriate convergence of a series handling the algorithm's error, discuss the 29 -> 5, etc. . This allows to install the formal decorum.